# Discovery of Novel STING Inhibitors Based on the Structure of the Mouse STING Agonist DMXAA

**DOI:** 10.3390/molecules28072906

**Published:** 2023-03-23

**Authors:** Jiajia Chang, Shi Hou, Xinlin Yan, Wei Li, Junhai Xiao

**Affiliations:** National Engineering Research Center for the Emergency Drug, Beijing Institute of Pharmacology and Toxicology, Beijing 100850, China

**Keywords:** STING, DMXAA, species selectivity, STING agonist, STING inhibitor

## Abstract

The stimulator-of-interferon-gene (STING) protein is involved in innate immunity. The drug DMXAA (5,6-dimethylxanthenone-4-acetic acid) proved to be a potent murine-STING (mSTING) agonist but had little effect on human-STING (hSTING). In this paper, we draw upon the comparison of different crystal structures and protein-ligand interaction relationships analysis to venture the hypothesis that the drug design of DMXAA variants has the potential to convert STING agonists to inhibitors. Based on our previous discovery of two DMXAA analogs, **3** and **4** (both could bind to STING), we structurally optimized them and synthesized new derivatives, respectively. In binding assays, we found compounds **11** and **27** to represent STING binders that were superior to the original structures and discussed the structure-activity relationships. All target compounds were inactive in cellular assays for the screening of STING agonistic activity. Gratifyingly, we identified **11** and **27** as STING inhibitors with micromolar activity in both hSTING and mSTING pathways. In addition, **11** and **27** inhibited the induction of interferon and inflammatory cytokines activated by 2′3′-cGAMP without apparent cytotoxicity. These findings break the rigid thinking that DMXAA provides the structural basis specifically for STING agonists and open up more possibilities for developing novel STING agonists or inhibitors.

## 1. Introduction

The stimulator-of-interferon-gene (STING) is an essential signaling molecule for intrinsic immunity, primarily mediating cytoplasmic DNA-induced natural immune responses [1]. When the DNA receptor cyclic guanosine-adenosine phosphate synthase (cGAS) detects intracellular double-stranded DNA, cGAS catalyzes the synthesis of the 2′,3′-cGAMP (Figure 1), a cyclic dinucleotide (CDN) capable of directly binding to and activating STING on the endoplasmic reticulum [2,3,4]. Upon activation, STING forms aggregates that are recruited downstream to TANK-binding kinase 1 (TBK1) and bind interferon regulatory factor 3 (IRF3), which phosphorylates IRF3, activates IRF3 dimers into the nucleus, induces type I interferon (IFN) expression and initiates an interferon immune response [1]. Many studies have shown that STING is involved in the pathogenesis of multiple diseases and that stimulation of STING induces effective immune responses to pathogenic infections and cancers; however, failure to modulate chronic inflammatory signaling leads to autoimmune and inflammatory diseases [5,6,7,8]. Due to the fundamental role of STING in the regulation of innate immunity, a large number of teams have begun to develop STING agonists or inhibitors.

Research on first-generation STING agonists focused on structural modification of CDN analogs to circumvent some limitations of endogenous CDNs, such as poor membrane permeability and susceptibility to hydrolysis by phosphatases [9,10]. The small molecule STING agonist 5,6-dimethylxanthenone-4-acetic acid (DMXAA), which has shown therapeutic promise against solid tumors in mouse models, could counteract the drawbacks of CDN derivatives [11]. However, the drug failed in human clinical trials [12]. It is further affirmed that DMXAA selectively binds to murine STING (mSTING) rather than to human STING (hSTING) [13,14], which impedes the therapeutic potential of DMXAA in humans. 10-Carboxymethyl-9-acridinone (CMA) is another representative murine-specific STING agonist used as a potent type I IFN inducer for antiviral therapy in the early 1970s [15]. Therefore, species selectivity is a vital factor in the development of small-molecule STING agonists. Substantial novel-skeleton hSTING agonists have been brought into view by the tireless efforts of researchers, including diABZI, SR717, MSA02, and others [16,17,18,19].

Increasing evidence shows that STING hyperactivation is associated with autoinflammatory and autoimmune diseases. It is necessary to limit the excessive activation of the STING signaling pathway, which highlights the potential value of STING inhibitors. In contrast to STING agonists, the development of STING inhibitors is still in its infancy, however, and no drug candidates have entered clinical studies yet. The known STING inhibitors include two types of compound: competitive antagonists (SN-011 and Merck-18) and covalent inhibitors (H151) [20,21,22]. The earlier identified covalent STING inhibitors interact with Cys88/91 or Cys91 which is located in the N-terminal transmembrane domain of STING outside the CDN-binding pocket [22]. Recently, two teams have successively reported two structural types of STING antagonists that occupy the 2′3′-cGAMP binding pocket to inhibit the activation of 2′3′-cGAMP, suggesting a double-edged effect in the CDN-binding pocket [20,21].

Here, we provide a new perspective. We suggest that the bottom pocket where DMXAA and CMA are located facilitates the development of STING inhibitors based on our deep data mining and comparison of the co-crystal structure information of the identified STING proteins. Our previous study reported a series of CMA and DMXAA analogs and identified compounds **3** and **4** which bind to hSTING but with weak cellular potency [23] (Figure 2). 

In this paper, regarding the design of STING inhibitors to discover more potent binding agents with more contacts in the bottom pocket, we performed structural optimizations for **3** and **4**, respectively. Subsequently, we conducted an SAR study based on competition binding assays. In bioactivity evaluation, compounds **11** and **27** acted as broad-spectrum binders and exhibited micromolar levels of STING inhibitory activity in multiple reported cells. In terms of docking structure, two compound **11** molecules hold hSTING in an inactive “open” conformation, thus competitively inhibiting endogenous 2′3′-cGAMP binding, which is similar to the crystal structure of Merck-18 and the docking structure of SN-011 [20,21].

## 2. Results

### 2.1. Design Strategy of DMXAA Derivatives as STING Inhibitors

#### 2.1.1. Identification of Design Sites for STING Inhibitors by Comparing Different Crystal Structures

The apo-protein of the STING ligand binding domain (LBD) crystallized as a symmetrical dimer in the non-ligand, which was proven not to be caused by the gel filtration and analytical ultracentrifugation analysis [24,25,26]. The STING dimer adopts a butterfly-like conformation with the ligand binding site positioned at the interface between the two monomers. Firstly, this paper examines the infrastructural and activation mechanisms of hSTING and mSTING proteins, comparing the differences between them, as well as unveiling some intriguing insights. For the apo structure, the natural conformation of both proteins shows that the mSTING is more concentrated than the hSTING (Figure 3). Shih et al. employed molecular dynamics (MD) simulations to study the differences between hSTING and mSTING, showing that hSTING prefers an open-inactive conformation, and mSTING prefers a closed-active conformation even without a ligand bound [27].

When bound to 2′,3′-cGAMP, the conformational transition of the hSTING is from open (apo-protein, lateral distance ~57 Å) to closed (cGAMP-bound protein, lateral distance ~40 Å), whereas the opposite is true for mSTING (29 Å to 40 Å). The structure of apo-hSTING differs dramatically from that of apo-mSTING, but the structures of the STING complexed with 2′,3′-cGAMP overlap. DMXAA, CMA, and cGAMP are STING agonists with diverse structural scaffolds, but their activation conformations upon binding to mSTING proteins are also essentially the same (Figure 3). The scenarios described above testify that the stimulation mechanisms of hSTING and mSTING are strikingly divergent; the conformations activated by STING are, however, relatively constant, independent of species selectivity and agonist backbone type (while limited to STING agonists that bring about conformational changes). The binding pocket volumes of the different crystal complexes are generally convergent (~300 Å³, Appendix A), which is further evidence of the stability of the STING protein activation conformation.

Secondly, based on the above findings of activation conformational stability, we superimposed co-crystal structures of the cGAMP, DMXAA, CMA, SR717, and MSA02, which are closed-conformational STING agonists, to explore the position of the different structural agonists in the STING protein binding sites. The protein pocket can partition into two regions: the bottom pocket and the top pocket, following the distribution of STING agonists in the protein pocket (Figure 4a). Except for mSTING agonists DMXAA and CMA being entirely at the bottom pocket, the other agonists could activate the hSTING pathway and lie in the upper region. We then mapped the interactions of the five STING agonists with amino acid residues bound to the STING protein (within 5 Å of the ligand; Appendix A), showing that the primary amino acids included R238, Y167, R232, S162, T263, and T267 (the mSTING corresponding residues sequence number minus one). Matching the residues sequentially to the pockets showed that T267, T263, and S162 lie in the bottom pocket; the top pocket contains R238, R232, and Y167 (Figure 4b). As reported, R238, Y167, and R232 are essential for agonist binding to STING, especially R238 (the mSTING protein corresponds to R237), the mutation of which would result in a complete loss of stabilization (ligand binding) [28,29,30]. Che et al. also revealed that the key-residue R238 dominates the binding of DMXAA, and the point mutations (S162A/E260I) can enhance the interaction of R238 with DMXAA by MD simulations [31]. Invariably, all of these residues belong to the upper region, which is the main factor contributing to the hSTING activity of the compounds in the top pocket.

To occupy the mSTING protein pocket, DMXAA or CMA adopted a unique binding mode in which two small-molecule agonists bind to a single mSTING homodimer. The DMXAAs firmly mount at the bottom pocket by hydrogen bonds between the keto group and the T266 side chain (hSTING corresponds to 267), while the carboxylate group interacts with the R237 and T262 side chains (Figure 5a). The interaction of DMXAA with R237 (the only key amino acid in the top pocket) is crucial for mSTING activity. An oft-overlooked detail is that DMXAA can only act on R237 of symmetric monomeric proteins, for example, DMXAA (molecule A) affecting R237B (Figure 5a). Because DMXAA sits at the bottom, the distance between molecule A and R237B is 3.06 Å, while the distance to R237A is greater than 5 Å (Figure 5b). Benefiting from the more compact pocket of the mSTING protein, DMXAA interacts effortlessly with R237 of the symmetric mSTING monomer but there is no way for DMXAA to associate with any of the R238 in the conformationally open apo-hSTING protein (Figure 3). Hence, the pocket position of DMXAA impairs the stimulation of the hSTING pathway. Recently, Merck, inspired by the 2:1 binding ratio of DMXAA, has discovered STING antagonists capable of occupying the binding pocket [21]. The crystal complex of Merck 18 and hSTING protein shows that the compound is mainly located below the protein pocket and interacts with S162, T263, and T267, all of which match the properties of DMXAA (Figure 5c). From this, hypotheses were generated: the top pocket is an excellent cavity for designing STING agonists; the bottom pocket is a more suitable nest for inhibitors, and the structure of DMXAA or CMA is optimized to have the potential to become a STING antagonist.

#### 2.1.2. Design of Novel Derivatives Based on Previous Research

Although mSTING and hSTING have high sequence identity (68% amino acid identity and 81% similarity) [32], DMXAA activates mSTING but has no effect on hSTING. Gao et al. reported that Q266I and S162A binding pocket mutations, coupled with the G230I lid substitution, rendered hSTING sensitive to DMXAA (Figure 6a) [33]. The researchers proposed that these DMXAA derivatives may have the potential to restore sensitivity to hSTING by modifying the C1/C2 (S162A substitution) and C7 (Q266I substitution) positions within the DMXAA ring containing polar groups (Figure 6b) [33]. Despite extensive studies on the mechanism of action and structural optimization of DMXAA [27,33], most analogs still fail to activate the human STING pathway and have reduced murine activity [14,34]. In a previous study, inspired by the point mutation hSTING can bind to DMXAA, our team also targeted DMXAA and CMA-like compounds for structural modifications and successfully identified compounds **3** and **4**, which have weakly agonistic activities in both hSTING and mSTING pathways [23]. We also discovered that the agonist activities of compounds **3** and **4** did not match the binding potencies (>10-fold shift), which was the foundation for our proposed inhibitor hypothesis (Figure 2).

We found that the inability of DMXAA to activate the hSTING pathway was related to the location of DMXAA in the bottom pocket (according to Section 2.1.1), which also provided us with the design idea of STING inhibitors: increasing the contacts of DMXAA derivatives with the bottom region through structural modifications to keep STING in a non-activated conformation. S162 and Q266 lie appropriately at the bottom of the binding site, so DMXAAs can be well-positioned in direct contact with both, facilitating the exploration of the effect of bottom pocket amino acids on ligands. The structure−activity relationships (SARs) analysis revealed that the 5,6-dimethyl moiety and the C7-position methoxy were crucial for the biological activity of **3** and **4**. As shown in Figure 6b, the 5,6-dimethyl group created hydrophobic interactions with multiple amino acids in the bottom pocket; due to the proximity of the C7 position of DMXAA to amino acid Q266, the corresponding methoxy modification allowed **3** and **4** to bind to hSTING. Therefore, our structural elaborations presupposed the locking of 5,6-dimethyl and C7 methoxy, and the design of novel derivatives focused on modifying the polar group at the C1/C2 position (affecting S162) and changing the carboxylic acid group (extending structural diversity). We designed and synthesized a series of derivatives using compounds **3** and **4** as scaffolds, thus validating our proposed inhibitor conjecture.

### 2.2. Chemistry

The preparation of all intermediate and target compounds is outlined in Figure 1. 1-methoxy-2,3-dimethyl-4-nitrobenzene (**5**) and 2-bromobenzoic acid (**7**) were commercially available. Initially, 1-methoxy-2,3-dimethyl-4-nitrobenzene was converted to 4-methoxy-2,3-dimethylaniline (**6**) via reduction with iron powder. Then, **6** and **7** were subjected to Ullmann reaction with potassium carbonate with catalysis by copper and copper (I) oxide in *N,N*-dimethylformamide to provide the corresponding intermediate **8**. The intramolecular condensation reaction of **8** was performed with Eaton’s reagent to afford intermediate 2-methoxy-3,4-dimethylacridin-9(10*H*)-one (**9**). The subsequent substitution of **9** was performed with ethyl bromoacetate, and the loss of 1 equiv of HBr afforded ethyl 2-(2-methoxy-3,4-dimethyl-9-oxoacridin-10(9*H*)- yl)acetate (**10**). Finally, ethyl acetate in **10** was hydrolyzed to carboxylic acid by sodium hydroxide yielding 2-(2-methoxy-3,4-dimethyl-9-oxoacridin-10(9*H*)- yl)acetic acid (**11**).

For the subsequent synthesis of more target compounds, we must first synthesize compound 2-amino-5-methoxy-3,4-dimethylbenzoic acid (**14**). **6** was reacted with chloral hydrate and hydroxylamine, forming acetamide and hydroxyamino, with the intermediate (*E*)-2-(hydroxyimino)-*N*-(4-methoxy-2,3-dimethylphenyl)acetamide (**12**). The subsequent Beckmann rearrangement reaction of the hydroxyamino of **12** was performed with sulfuric acid to afford lactam derivative **13**. Finally, intermediate **13** was hydrolyzed with hydrogen peroxide and sodium hydroxide in ethanol solution affording **14**. Using **14** as the starting material for the reaction, we further synthesized a series of 8-methoxy-9,10-dimethyl-6*H*-pyrrolo[3,2,1-*de*]acridine-1,6(2*H*)-dione derivatives with substituents at R^1^ and R^2^. Initially, **14** and **15–19** were subjected to Ullmann reaction with potassium carbonate with catalysis by copper and copper(I) oxide in *N,N*-dimethylformamide to provide corresponding intermediates **20–24**. The intramolecular condensation reactions of **20–24** were performed with Eaton’s reagent to afford 8-methoxy-9,10-dimethyl-6*H*-pyrrolo[3,2,1-*de*]acridine-1,6(2*H*)-dione derivatives (**25–29**). Similarly, using 14 as the starting material for the reaction, we further synthesized a series of 7-methoxy-5,6-dimethyl-9-oxo-9,10-dihydroacridine-4-carboxylic acid derivatives with substituents at R. Initially, **14** and **7** or **30** were subjected to Ullmann reaction with potassium carbonate with catalysis by copper and copper(I) oxide in *N,N*-dimethylformamide to provide corresponding intermediates **31** and **32**. The intramolecular condensation reactions of **31** and **32** were performed with Eaton’s reagent to afford 7-methoxy-5,6-dimethyl-9-oxo-9,10-dihydroacridine-4-carboxylic acid derivatives (**33**,**34**). Thus, we designed, synthesized, and screened 16 acridone analogs, whose structures are summarized in Table 1.

### 2.3. Binding Potencies of New Compounds to STING and their SARs

To directly confirm the effectiveness of the structural modification, we first screened all the synthesized compounds by cGAMP displacement assays. In addition to species selectivity, there are five variants of STING that are polymorphic in humans, R232 (WT, 58% of the population), HAQ (20%), H232 (13%), AQ (7%), and Q (2%) [32]. Hence, we used the commercial competition assay kits by homogeneous time-resolved fluorescence (HTRF) technology to test the biochemical potency of new compounds on mSTING and various hSTING isoforms (WT, H232, and AQ) and compared them to the control. As anticipated, control compounds **1–4** were all bound to mSTING; from the hSTING screen, compounds **3** and **4** had IC_50_ values at the micromolar level in all three displacement assays. Interestingly, some optimization compounds have better activity with broad-spectrum effects; we show SARs based on their activity data in Table 2.

First, based on compound **3**, a halogen atom or methoxy group was introduced at the C1/C2 site (Table 1). Compound **27**, as the optimal derivative of compound **3** (C1 position substituted with F), has better overall activity than compound **3** (single-digit micromolar activity in all biochemical assays, Table 2). The R_1_-modified compound **25** has a wide-spectrum binding effect but is not as active as the F-substituted or unsubstituted version. Compared to compound **3**, the compounds (**26**, **28**, and **29**) with the C2-site introduced group all showed different degrees of decrease in binding affinity, especially compound **29**. Second, keeping the backbone of compound **4** and replacing the halogen atom against the C1/C2 site, we obtained compounds **35–38** (Table 1). These compounds were inactive in all four displacement assays at concentrations up to 100 μM, demonstrating that substitution with a halogen atom at the C1/C2 site is not tolerated. Then, we turned the acetate group on compound **4** into a carboxyl group (**33**) and introduced methoxy at the C2 site (**34**). Unfortunately, this change was a failure since both compounds remained inactive. Learning from our failure, we further explored the SARs of the compound **4** derivatives by shifting the acetate group to the N position and leaving the C1/C2 positions unmodified. Compound **11** shows binding to a wide range of STING variants, all with IC_50_s below 20 μM; the activity is superior to compound **4** and comparable to compound **27**. In addition, compound **9** (no carboxyl group at the N site) and compound **10** (N-acetate ethyl) are synthetic precursors of compound **11** that cannot bind to STING, indirectly demonstrating the importance of the carboxylic acid group.

### 2.4. Cellular Biological Evaluation 

#### 2.4.1. In Vitro Screening of New Analogs with STING Agonist Activities

We systematically screened all compounds for STING agonist activity using the 293T-hSTING-WT, 293T-mSTING and THP1-KO-STING reporter cell lines (provided by InvivoGen). Using 293T-hSTING-R232 cells and 293T-mSTING cells, we showed activation of the IRF pathway as an indirect measure of type I IFN induction by monitoring secretory embryonic alkaline phosphatase (SEAP) activity, contributing to our study of the species selectivity of the compounds. THP1-KO-STING cells were generated from THP1-Dual cells by stable knockout of the STING gene, and we employed THP1-KO-STING cells to confirm whether the compound exhibits the function of STING-dependent cytokine induction. We compared the STING agonist activities of all the targeted derivatives to those of the reference agonists for murine (DMXAA) and human (2′,3′-cGAMP) STING. Surprisingly, none of the synthesized compounds could activate either the hSTING or mSTING pathways (maximum concentration at 200 μM), which is seriously inconsistent with the results of the binding assays. Therefore, our screened STING binding agents may have potential binding patterns as novel STING antagonists.

#### 2.4.2. In Vitro Screening of STING Binders with STING Inhibitor Activities

First, we did pre-experiments: with the 1 μM covalent inhibitor H151 as a positive reference, we explored its inhibitory levels in mSTING and hSTING reporter cell lines co-cultured with different concentrations of 2′3′-cGAMP to determine the optimal conditions for screening the inhibitor (see Section 4.2.3 for protocol details). Second, we initially screened the inhibitory potency of the new compounds by treatment with the pre-experimental methods at a concentration of 100 μM. We found that the broad-spectrum STING-binding compounds **11** and **27** exhibited excellent inhibition in both murine and human STING reporter cells, with compounds **3** and **4** and the other compounds not being as effective. Third, we set concentration gradients for compounds **11** and **27** to test both IC_50_ values and compare them with H151. Compounds **11** and **27** were active at micromolar concentrations inhibiting 2′3′-cGAMP-induced IFN-β expression in both hSTING and mSTING pathways. 

With IC_50_ values, compound **11** (hSTING 19.93 μM and mSTING 15.47 μM) outperformed compound **27** (hSTING 38.75 μM and mSTING 30.81 μM) (Figure 7a). In comparison, the IC_50_ values of H-151 in 293T-hSTING and 293T-mSTING cells were 1.04 and 0.82 μM (Figure 7a), indicating that the covalent inhibitor H-151 exhibits a better inhibitory effect on STING-dependent signaling in cell-based assays. To further confirm the inhibitory property of compounds **11** and **27**, we employed another THP-1 human monocyte reporter cell line THP1-Dual-hSTING-R232, which uses a dual-reporter system to report on IRF activation as an indirect measure of type I IFN induction and on NF-κB activation as an indirect measure of proinflammatory cytokine induction. After the THP-1 reporter cell line stimulation with 2′,3′-cGAMP, we added STING inhibitors with appropriate concentrations for inhibiting the inductions of I IFN and proinflammatory cytokine. As anticipated, compound **11**, **27**, and H151 significantly inhibited STING-triggered IRF and NF-κB pathway activations (Figure 7b). In addition, to exclude the effect of cytotoxicity of the compounds on the inhibitory activity, we used the CellTiter-Glo kit to assay cellular activity. Gratifyingly, compounds **11** and **27** showed no evidence of cytotoxicity to 293T and THP-1 cells by cell viability assay when added at concentrations (5 to 100 μM) that inhibit STING, in contrast to H151, which already showed significant cytotoxicity at 10 μM (Figure 7c). These studies identified compounds **11** and **27** as moderate STING inhibitors, breaking species sensitivity and inhibiting the IRF and NF-κB dual pathways without apparent cytotoxicity.

### 2.5. The Structural Basis of the Activity of Compound 11 Was Investigated by Docking

The mechanism of competitive STING antagonists is to occupy the binding site and disrupt the activation conformation of the STING protein [20,21]. Specific to hSTING, antagonists leave the hSTING dimer in an “open” conformation. We used the Glide docking to determine the interaction between human STING CTD (PDB ID code 6MXE) and the best active compound **11**. In the docking structure, the two compounds **11** are partially parallel to each other and lie at the bottom of the cleft of the hSTING dimer (Figure 8a), locking hSTING in an inactive open conformation (Appendix A). Compound **11** produces mainly hydrophobic interactions, with the tricyclic structure of the ligand allowing more contact with the interface in the bottom pocket. As with DMXAA or CMA, the carboxyl group forms a hydrogen bond with the side chain of T263, while the keto group forms a hydrogen bond with the side chain of T267. The methoxy at position C2 interacts intimately with S162, resulting in ligand-ligand vicinity towards each other. The bis-methyl substituents not only generate ligand-ligand interactions but are also conveniently located at the entrance to the top pocket, which prevents the binding of natural STING agonists to key residues (Figure 8b).

## 3. Discussion

In light of the extensive reports on the mechanism and crystal structure of STING, we have the opportunity to explain the scientific problem of why DMXAA does not possess human STING agonistic activity. In this study, by comparison of various STING crystal structures, we obtained some interesting findings: 1. The STING activation conformation is stable; 2. There are different activation mechanisms for hSTING and mSTING; 3. DMXAA and CMA are at the bottom of the binding site distinguishing them from other agonists. Given the different activation mechanisms of mSTING and hSTING, the mSTING agonists DMXAA or CMA located in the bottom pocket cannot exert an activation in hSTING because apo-hSTING has a larger binding cavity preventing the ligand from associating with R238.

Gao et al. reported that single mutations (G230I, S162A and Q266I) endow hSTING with the same DMXAA sensitivity as mSTING [33]. MD simulations revealed that the lid region mutation G230I side chain is sufficient to form a steric barrier to prevent DMXAA excretion, whereas DMXAA readily exits in hSTING WT [27]. Based on the structural modification corresponding to hSTING point mutations (S162A/Q266I) in the binding site, substantial DMXAA and CMA analogs have been available but no potent hSTING agonists. The reason behind this might be that the subtle structural discrepancy between “natural” and “mutated” hSTINGs might play a crucial role in the recognition process. Through MD simulations, Che et al. found that the unnaturally mutated hSTINGs disturb the coordinated motions of water molecules and alter the amount of water expelled upon ligand binding, which is more conducive to restore DMXAA activation of hSTING [31]. Encouraged by the study discovery of STING inhibitors at the bottom binding site [21], we venture to envisage that DMXAA, located in the same region, has the potential to be transformed into novel STING inhibitors. In a previous study, we fused the structures of DMXAA and CMA with subtle structural modifications and successfully discovered **3** and **4** that could bind to hSTING, but their weak STING agonistic activity does not match their binding potency. In this paper, the design idea is to further enhance the contacts of DMXAA analogs with the bottom pocket to competitively resist the binding of 2′,3′-cGAMP, so we designed and synthesized optimized structures of compounds **3** and **4**.

The first screening step was to test the biochemical activity using the binding kits of multiple STING variants to characterize the binding potency of the novel derivatives. The results of the binding assays proved the soundness of our design directions; we identified **11** and **27** as broad-spectrum STING binders (superior to **3** and **4**) and discussed SARs. For the analogs of compound **3**, the modifications at the C1 position significantly improved the binding to various STING variants, while the C2-modified compounds had no effect. However, compound **4** was not well tolerated for structural modifications in C1/C2. After relocating the carboxylic acid position (C4 to N), we found the better STING binder **11** and determined that the carboxylic acid group was essential for activity. Next, we screened all compounds for cellular-level activity using three reporter cell lines, and none of the target compounds were active. We then established a screening method for STING inhibitors and rescreened all new analogs. STING binding agents **11** and **27** exhibited micromolar inhibitory activities, with data comparable to the competitive binding assay. The covalent inhibitor H151 showed slightly better STING inhibitory activity than **11** and **27** in vitro. In addition, we further validated that **11** and **27** can effectively inhibit 2′,3′-cGAMP-induced activation of STING dual pathways. Most impressively, **11** and **27** maintain a high level of cell viability at high concentrations, which is not possible with H151. Cytotoxicity is a common problem with covalent inhibitors, and limits the application of H151.

By docking analysis, we have acquired insight into the structural basis of best active compound **11** activity. Compound **11** occupies the bottom pocket perfectly, holding hSTING in an inactive open conformation and thus competitively inhibiting 2′,3′-cGAMP binding. The structure of the tricyclic ring of acridinone is key to generating hydrophobic interactions, with the carboxylic acid group and ketone moiety anchoring the compound at the bottom of the pocket. We speculate that the methoxy produces intimate associations with the S162, pulling the protein conformation closed marginally but not enough to turn it into an agonistic conformation. The bis-methyl structure creates a spatial site-blocking that enhances ligand-ligand interactions and detracts from agonist entry into the top pocket. Currently, there are no reported co-crystal structures of mSTING and STING antagonists, so we cannot explain the mechanism of action of compound **11** with mSTING using a docking approach. However, we conjecture that there are two possibilities: one is that the acting principle is identical to that of hSTING; the other is that compound **11** keeps mSTING in a more aggregated apo conformation and 2′,3′-cGAMP cannot enter the pocket to exert the agonistic effect.

In summary, our subtle structural optimizations of the weaker STING agonists have successfully reversed the biological function to discover STING inhibitors. These findings illustrate the complexity of the STING binding pockets and provide new research insights for the drug development of STING agonists or inhibitors.

## 4. Materials and Methods

### 4.1. Chemistry

Solvents and reagents used in the synthesis were obtained from Beijing Innochem Science and Technology Co., Ltd. (Beijing, China). The structures of the products were identified by ^1^H and ^13^C-NMR spectroscopy (JNM-ECA-400, Japan). The molecular weights of the products were measured by high resolution mass spectrometry (HRMS) with electrospray ionization (ESI) as the ionization mode (Agilent 1260-G6230A, Germany). NMR and mass spectra of the compounds are provided in the Appendix A. Compounds **35–38** were synthesized and identified by us; please refer to the previously reported article for the details [23].

Ethyl 2-(2-methoxy-3,4-dimethyl-9-oxoacridin-10 (9*H*)-yl)acetate (**10**): To a solution of 4-methoxy-2,3-dimethylaniline (**6**) (0.97 g, 6.4 mmol) and 2-bromo-benzoic acid (**7**) (1.29 g, 6.4 mmol) in DMF (6 mL) at room temperature, we subsequently added powdered Cu (0.05 g), Cu_2_O (0.05 g) and K_2_CO_3_ (0.71 g, 5.1 mmol). The reaction mixture was heated at 110 °C for 12 h. Upon removal of the solvents under a vacuum, the residue was dissolved in 1 N NaOH solution (25 mL). The crude product was obtained by precipitation upon acidification of the filtrate with conc. HCl. After drying, the crude product was added to Eaton’s reagent (5 mL) at room temperature, then the mixture was heated to 90 °C for 1 h. The cooled reaction mixture was dropped into saturated aqueous NaHCO_3_ solution. The precipitate was filtered to collect the rough product. Purification of the residue by silicagel column chromatography provided **9**, a pale yellow solid (0.61 g, 37.7% yield). A solution of NaH (1.43 mmol) and **9** (0.33 g, 1.3 mmol) in DMF (5 mL) at room temperature was stirred for 1 h and then cooled to 5–7 °C. The ethyl bromoacetate (0.43 g, 2.6 mmol) was added to the resulting mixture and continuously stirred at room temperature for 20 h. After completion of the reaction (TLC), the reaction mixture was poured into ice water (15 mL). The resulting precipitates were filtered off, dried, and then extracted with chloroform. Evaporation of the solvent gave a crude ester which was purified by recrystallization to provide **10**, a yellow solid (0.33 g, 76% yield). ^1^H NMR (400 MHz, DMSO-D6) δ 8.30 (d, *J* = 8.4 Hz, 1H), 8.08 (d, *J* = 8.6 Hz, 1H), 7.75–7.67 (m, 1H), 7.57–7.50 (m, 1H), 7.48 (s, 1H), 5.02 (s, 2H), 4.22 (q, *J* = 7.1 Hz, 2H), 3.96 (s, 3H), 2.77 (s, 3H), 2.34 (s, 3H), 1.21 (t, *J* = 7.1 Hz, 3H). ^13^C NMR (101 MHz, DMSO-D6) δ (ppm): 169.25, 157.87, 155.96, 147.46, 146.72, 135.46, 132.15, 130.17, 129.51, 125.88, 122.48, 119.80, 119.46, 95.26, 72.06, 61.28, 56.06, 14.59, 14.34, 13.77. HRMS (ESI) *m/z* [M+H]+ calculated for C_20_H_21_NO_4_: 339.3910 found: 340.1546.

2-(2-methoxy-3,4-dimethyl-9-oxoacridin-10(9*H*)-yl)acetic acid (**11**): A solution of **10** (0.37 g, 1.1 mmol) and NaOH (0.05 g, 1.3 mmol) in ethanol (20 mL) and water (2 mL) was heated to 60 °C for 1 h. Upon removal of the solvents, the resulting residue was dissolved in water and filtered. After neutralization of the filtrate with conc. HCl, the mixture was filtrated to get the crude solid, subsequently recrystallized by methanol to give **11**, a pale yellow power (0.29 g, 85.7% yield). ^1^H NMR (400 MHz, DMSO-D6) δ 10.51 (s, 1H), 8.33 (d, *J* = 6.9 Hz, 1H), 8.04 (d, *J* = 8.5 Hz, 1H), 7.69 (s, 1H), 7.67–7.62 (m, 1H), 7.45 (t, *J* = 6.8 Hz, 1H), 4.52 (s, 2H), 3.90 (s, 3H), 2.75 (s, 3H), 2.31 (s, 3H). ^13^C NMR (101 MHz, DMSO-D6) δ (ppm): 159.04, 155.63, 147.47, 146.64, 134.86, 131.52, 129.56, 128.98, 125.15, 122.85, 120.53, 119.69, 101.77, 96.02, 74.81, 55.69, 14.42, 13.33. HRMS (ESI) *m/z* [M+H]+ calculated for C_18_H_17_NO_4_: 311.3370 found: 312.1229.

5-chloro-8-methoxy-9,10-dimethyl-6*H*-pyrrolo [3,2,1-*de*]acridine-1,6(2*H*)-dione (**25**): To a solution of 6-carboxy-4-methoxy-2,3-dimethylbenzenaminium (**14**) (1.3 g, 6.4 mmol) and 2-(2-bromo-4-chlorophenyl)acetic acid (**15**) (1.6 g, 6.4 mmol) in DMF (6 mL) at room temperature, we subsequently added powdered Cu (0.05 g), Cu_2_O (0.05 g) and K_2_CO_3_ (0.71 g, 5.1 mmol). The reaction mixture was heated at 110 °C for 12 h. Upon removal of the solvents under a vacuum, the residue was dissolved in 1 N NaOH solution (25 mL). The crude product was obtained by precipitation upon acidification of the filtrate with conc. HCl. After drying, the crude product was added to Eaton’s reagent (5 mL) at room temperature, then the mixture was heated to 90 °C for 1 h. The cooled reaction mixture was dropped into saturated aqueous NaHCO_3_ solution. The precipitate was filtered to collect the rough product. Purification of the residue by silicagel column chromatography provided **25**, a yellow solid (0.80 g, 38.6% yield). ^1^H NMR (400 MHz, DMSO-D6) δ 8.17 (d, *J* =8.1, 1.5 Hz, 1H), 7.86 (d, *J* = 6.9 Hz, 1H), 7.51 (s, 1H), 3.84 (s, 3H), 3.78 (s, 2H), 2.49 (s, 3H), 2.27 (s, 3H); HRMS (ESI) *m/z* [M+H]+ calculated for C_18_H_14_ClNO_3_: 327.7640 found: 328.0734.

4-chloro-8-methoxy-9,10-dimethyl-6*H*-pyrrolo[3,2,1-*de*]acridine-1,6(2*H*)-dione (**26**): The synthesis of this compound was similar to **25**. **15** was replaced with 2-(2-bromo-5-chlorophenyl)acetic acid (**16**). We obtained 0.82 g **26** as a pale-yellow solid with a yield of 38.7%. ^1^H NMR (400 MHz, DMSO-D6) δ 7.73 (s, 1H), 7.52 (s, 1H), 7.38 (s, 1H), 3.71 (s, 3H), 3.64 (s, 2H), 2.36 (s, 3H), 2.13 (s, 3H); HRMS (ESI) *m/z* [M+H]+ calculated for C_18_H_14_ClNO_3_: 327.7640 found: 328.0734.

5-fluoro-8-methoxy-9,10-dimethyl-6*H*-pyrrolo[3,2,1-*de*]acridine-1,6(2*H*)-dione (**27**): The synthesis of this compound was similar to **25**. **15** was replaced with 2-(2-bromo-4-fluorophenyl)acetic acid (**17**). We obtained a deep yellow solid **27** with 0.84 g (42.1% yield). ^1^H NMR (400 MHz, DMSO-D6) δ 8.21 (d, *J* = 8.0 Hz, 1H), 7.68 (s, 1H), 7.23 (d, *J* = 7.5 Hz, 1H), 3.88 (s, 3H), 3.81 (s, 2H), 2.53 (s, 3H), 2.30 (s, 3H); HRMS (ESI) *m/z* [M+H]+ calculated for C_18_H_14_FNO_3_: 311.3124 found: 312.1030.

4-fluoro-8-methoxy-9,10-dimethyl-6*H*-pyrrolo[3,2,1-*de*]acridine-1,6(2*H*)-dione (**28**): The synthesis of this compound was similar to **25**. **15** was replaced with 2-(2-bromo-5-fluorophenyl)acetic acid (**18**). 0.81 g deep yellow solid was obtained (40.5% yield). ^1^H NMR (400 MHz, DMSO-D6) δ 7.86 (s, 1H), 7.64 (s, 1H), 7.51 (s, 1H), 3.83 (s, 3H), 3.77 (s, 2H), 2.49 (s, 3H), 2.26 (s, 3H); HRMS (ESI) *m/z* [M+H]+ calculated for C_18_H_14_FNO_3_: 311.3124 found: 312.1030.

4,8-dimethoxy-9,10-dimethyl-6*H*-pyrrolo[3,2,1-*de*]acridine-1,6(2*H*)-dione (**29**): The synthesis of this compound was similar to **25**. **15** was replaced with 2-(2-bromo-5-methoxyphenyl)acetic acid (**19**). 0.73 g yellow solid was provided with a 35.2% yield. ^1^H NMR (400 MHz, DMSO-D6) δ 7.93 (s, 1H), 7.61 (s, 1H), 7.47 (s, 1H), 3.84 (s, 3H), 3.81 (s, 3H), 3.67 (s, 2H), 2.44 (s, 3H), 2.28 (s, 3H); HRMS (ESI) *m/z* [M+H]+ calculated for C_19_H_17_NO_4_: 323.3480 found: 324.1230.

7-methoxy-5,6-dimethyl-9-oxo-9,10-dihydroacridine-4-carboxylic acid (**33**): To a solution of 6-carboxy-4-methoxy-2,3-dimethylbenzenaminium (**14**) (1.30 g, 6.4 mmol) and 2-bromo-benzoic acid (**7**) (1.29 g, 6.4 mmol) in DMF (6 mL) at room temperature, we subsequently added powdered Cu (0.05 g), Cu_2_O (0.05 g) and K_2_CO_3_ (0.71 g, 5.1 mmol). The reaction mixture was heated at 110 °C for 12 h. Upon removal of the solvents under a vacuum, the residue was dissolved in 1 N NaOH solution (25 mL). The crude product was obtained by precipitation upon acidification of the filtrate with conc. HCl. After drying, the crude product was added to Eaton’s reagent (5 mL) at room temperature, then the mixture was heated to 90 °C for 1 h. The cooled reaction mixture was dropped into saturated aqueous NaHCO_3_ solution. The precipitate was filtered to collect the rough product. Purification of the residue by silicagel column chromatography provided **33**, a pale yellow solid (1.00 g, 52.7% yield). ^1^H NMR (400 MHz, DMSO-D6) δ 13.00 (s, 1H), 10.28 (s, 1H), 8.01 (d, *J* = 9.7 Hz, 1H), 7.69 (d, *J* = 8.5 Hz, 1H), 7.48 (t, *J* = 9.0 Hz, 1H), 7.34 (s, 1H), 3.67 (s, 3H), 2.32 (s, 3H), 2.10 (s, 3H); HRMS (ESI) *m/z* [M+H]+ calculated for C_17_H_15_NO_4_: 297.3100 found: 298.1073.

2,7-dimethoxy-5,6-dimethyl-9-oxo-9,10-dihydroacridine-4-carboxylic acid (**34**): The synthesis of this compound was similar to **33**. The **7** was replaced with 2-bromo-5-methoxybenzoic acid (**30**). The yellow solid gave 1.05 g with a 50.2% yield. ^1^H NMR (400 MHz, DMSO-D6) δ 12.96 (s, 1H), 10.45 (s, 1H), 7.84 (s, 1H), 7.54 (s, 1H), 7.50 (s, 1H), 3.84 (s, 3H), 3.81 (s, 3H), 2.48 (s, 3H), 2.27 (s, 3H); HRMS (ESI) *m/z* [M+H]+ calculated for C_18_H_17_NO_5_: 327.3360 found: 328.1179

### 4.2. Biological Assay

Mouse STING binding kits, human STING WT binding kits, human AQ STING binding kits, human H232 STING binding kits were purchased from Cisbio. 293T mSTING (ISG/KI-IFNb) cells, 293T hSTING-R232 (ISG/KI-IFNb) cells, THP1-KI-hSTING-R232 cells, THP1-KO-STING cells were purchased from Invivogen. 293T mSTING cells and 293T hSTING-R232 cells were cultured using DMEM (Gibco, Waltham, MA, USA), 2 mM L-glutamine (Sigma-Aldrich, St. Louis, MO, USA), 4.5 g/L glucose (Sigma-Aldrich), 10% FBS (Gibco), Pen-Strep (100 U/mL-100 μM) (Gibco), 100 μM normocin (Invivogen, San Diego, CA, USA), and supplemented with selective antibiotics (Invivogen) blasticidin (10 µM), hygromycin (100 µM) and zeocin (100 µM). THP1-KO-STING cells and THP1-KI-hSTING-R232 cells were cultured using RPMI-1640 (Gibco), 2 mM L-glutamine (Sigma-Aldrich), 25 mM HEPES (Sigma-Aldrich), 10% heat-inactivated FBS (Gibco), Pen-Strep (100 U/mL-100 μM) (Gibco), 100 μM normocin (Invivogen), and supplemented with selective antibiotics (Invivogen) blasticidin (10 μM) and zeocin (100 μM). QUANTI-Blue solution, QUANTI-Luc, 2′,3′-cGAMP were purchased from Invivogen. H151 was obtained from Beijing Innochem Science and Technology Co., Ltd. (Beijing, China). CellTiter-Glo luminescent cell viability assay kit was obtained from Promega.

#### 4.2.1. HTRF STING Binding Competitive Assay

According to the instructions of STING binding kits, the following solutions were successively added to each well in the 384-well plate: 5 μL of the detection compounds or 2′,3′-cGAMP (positive control) with different concentrations or diluent (negative control); 5 μL of human STING 6His-tagged protein (negative control well was added to detection buffer); 10 μL of STING ligand d2 and Anti 6His-Tb3^+^ premixed working solution. After sealing the plate and incubation at room temperature for 3 h, the fluorescence values at 665 nm and 620 nm were read. The ratio of the two fluorescence intensities (665 nm/620 nm) was used to estimate the binding potency of compounds. The IC_50_ (50% inhibitory concentration) values were calculated using software GraphPad Prism.

#### 4.2.2. Cellular Assay for Screening of Compounds for the Agonistic Activity

180 μL amounts of 293T mSTING cells, 293T hSTING-R232 cells and THP1-KO-STING cells suspension were distributed in 96-well flat-bottom plates with density of ~50,000 cells/well (293T) or ~100,000 cells/well (THP1). An amount of 20 μL of either saline or a saline solution of a test compound, or 2′,3′-cGAMP as positive reference was added, and the cells were incubated at 37 °C with 5% CO_2_ for 48 h. Subsequently, an amount of 20 μL of supernatant was added into a 96-well flat-bottom plate, followed by 180 μL of QUANTI-Blue solution to each well. The plate was incubated at 37 °C for 3 h, and SEAP levels (the IRF pathway activity) were determined using a spectrophotometer at 620–655 nm.

#### 4.2.3. Cellular Assay for Screening of Compounds for Inhibitory Activity

Pre-experiment: 180 μL of 293T mSTING cells, 293T hSTING-R232 cells (~50,000 cells/well) were stimulated for 48 h at 37 °C in a 5% CO_2_ incubator with 10 μL 2′,3′-cGAMP (3.125 µM, 6.25 µM) and 10 μL H151 (1.0 µM). Subsequently, an amount of 20 μL of supernatant was added into a 96-well flat-bottom plate, followed by 180 μL of QUANTI-Blue solution to each well. The plate was incubated at 37 °C for 3 h, and SEAP levels (the IRF pathway activity) were determined using a spectrophotometer at 620–655 nm.

180 μL of 293T mSTING cells, 293T hSTING-R232 cells suspension was distributed in 96-well flat-bottom plates with density of ~50,000 cells/well. An amount of 10 μL 2′,3′-cGAMP (3.125 µM) and compound **11** or compound **27** (0.78 µM, 1.56 µM, 3.13 µM, 6.25 µM, 12.5 µM, 25 µM, 50 µM, 100 µM) or H151 (0.08 µM, 0.16 µM, 0.31 µM, 0.63 µM, 1.25 µM, 2.5 µM, 5 µM) were added, and the cells were incubated at 37 °C with 5% CO_2_ for 48 h. Subsequently, an amount of 20 μL of supernatant was added into a 96-well flat-bottom plate, followed by 180 μL of QUANTI-Blue solution to each well. The plate was incubated at 37 °C for 3 h, and SEAP levels (the IRF pathway activity) were determined using a spectrophotometer at 620–655 nm. IC_50_ values were calculated by GraphPad software.

#### 4.2.4. Cellular Assay for Inhibition of STING Dual Pathways

180 μL of THP1-KI-hSTING-R232 cell suspension was distributed in 96-well flat-bottom plates with a density of ~100,000 cells/well. An amount of 10 μL 2′,3′-cGAMP and 10 μL compound **11** (20 µM) or compound **27** (33 µM) or H151 (2 µM) was added, and the cells were incubated at 37 °C with 5% CO_2_ for 24 h. Subsequently, an amount of supernatant was added into a 96-well white (opaque) plate, followed by QUANTI-Luc or QUANTI-Blue solution to each well and luminescence levels or SEAP levels were read according to the manufacturer’s instructions. This allows the simultaneous study of the IFN regulatory factor (IRF) pathway, by assessing the activity of Lucia luciferase and the NF-κB pathway, by monitoring the activity of SEAP.

#### 4.2.5. CellTiter-Glo Luminescent Cell Viability Assay

Cell viability was measured by CellTiter-Glo luminescent cell viability assay kit according to the manufacturer’s instructions. Briefly, cells were incubated with three different concentrations (5 µM, 10 µM, 100 µM) of compound **11** or compound **27** or H-151 for 48 h. 100 μL CellTiter-Glo reagent was added into 96-well assay plate for 10 min, then luminescence was recorded.

### 4.3. Molecular Docking of Compound **11**

The crystal structure of STING complex (PDB ID: 6MXE) was taken from a Protein Data Bank entry and used as the starting point. Protein was prepared using the Protein preparation of Maestro and split into chain A and chain B. We generated a binding site in the LBD of STING monomer based on the original ligand (Merck-18) using Receptor Grid Generation. We used the SP precision of Glide-docking for the molecular docking part, allowing compound **11** to generate at most 20 poses. We ultimately produced 16 bound conformations (all in the bottom pocket), and tested the binding free energy of the complexes using the Prime MM-GBSA module in Schrödinger’s software. Furthermore, based on the docking score, glide emodel score, and MMGBSA dG Bind score, we selected the optimal conformation scored first in two and ranked third in one (seen in Appendix A). Finally, we merged the best conformations of the A and B chains to obtain the complete docking structure.

## Data Availability

All data are contained within the article or supplementary material. The numerical data represented in the Figures are available on request from the corresponding author.

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
