# Peer review of "Discovery of Novel STING Inhibitors Based on the Structure of the Mouse STING Agonist DMXAA"

_molecules, 2023, doi:10.3390/molecules28072906_

Round 1

Reviewer 1 Report

In the manuscript, the authors analyzed the crystal structures of murine and human STING proteins and rationalized why DMXAA is a potent murine-STING (mSTING) agonist but remains insensitive to human-STING (hSTING). According to this analysis, they synthesized a number of analogs of DMXAA and identified two of them as STING inhibitors with micromolar activity in both the hSTING and mSTING pathways.
The manuscript is written clearly and will be interesting to the readers of the Molecules journal.
There are only some minor corrections required (shown in the annotated manuscript):

Line ca. 75: The structures of 3 and 4 should be shown somewhere here. Page 7 is too far.

Lines 80-84: To be deleted. It is repetition of the information from the abstract.

Line 236: “we must firstly synthesized” ® “we must first synthesize”

Lines 239-248: correct the chemical names (italics required, superfluous spaces to be deleted)

Table 1 (optional): Highlighting the best compounds would be advantageous for the readers.

Table 2 (optional): Highlighting the best results would be advantageous for the readers.

Line 479 and further: correct the chemical names (italics required, superfluous spaces to be deleted)

References: a number of references are corrupted (no journal name, etc.), they must be corrected

Author Response

We are very grateful for your recognition of our work and admire your rigorous academic attitude. Based on your annotations, we have made a complete revision of the manuscript (manuscript ID: molecules-2278643), which is indeed more reader-friendly. Thank you again for your valuable comments. We have marked the changes in red in the revised manuscript (please see the attachment) and hope to receive your approval.

The correction is as follows:

  1. We have placed the structure figure of compounds 3 and 4 on page 3.
  2. Removed duplicate content in lines 80-84.
  3. Fixed language issues and incorrect chemical names.
  4. Highlighted the best results in Tables 1 and 2.
  5. Missing and inconsistent references have been completely corrected.

Reviewer 2 Report

The manuscript “Discovery of novel STING inhibitors based on the structure of
the mouse STING agonist DMXAA”
analyzes ligand-protein interactions from X-ray structures to identify new STING inhibitors. The addressed topic is  interesting and the results look promising.

I have several minor observations:

-        It would be interesting to see the differences between them hSTING and mSTING stimulation mechanisms using molecular dynamics; also the binding of compound 11

-        Figure 3 - if possible, please use colors that allow a better contrast between cGAMP and SR71 (green and cyan)

Author Response

We thank you much for providing constructive feedback. We have revised our manuscript (Manuscript ID: molecules-2278643) according to your comments and added new analyses to strengthen our work. The main revisions and new additions we have undertaken are summarized below and discussed in detail in the point-by-point responses.

Point 1: It would be interesting to see the differences between the hSTING and mSTING stimulation mechanisms using molecular dynamics.

Response 1: Thank the reviewer for your valuable comment. Due to our oversight, we did not mention that some teams have studied differences between the hSTING and mSTING stimulation mechanisms using molecular dynamics. We carefully researched and found that parts of our manuscript coincided with these studies, so we added MD validation findings from the previous articles in our paper, which is indeed more reader-friendly. The revised parts of the manuscript are marked in red, and more relevant contents are summarized as follows:

Shih et al. employed molecular dynamics simulations to investigate the differences between hSTING and mSTING that may affect DMXAA binding (DOI: 10.1016/j.bpj.2017.10.027). First, they compared hSTING G230I with WT, a single lid region mutation G230I in hSTING (the corresponding residue in mSTING is an Ile) that restores the sensitivity to DMXAA. Simulations revealed that an Ile side chain is sufficient to form a steric barrier to prevent DMXAA excretion, whereas DMXAA readily exits in hSTING WT. Second, the results show that hSTING prefers an open-inactive conformation, and mSTING prefers a closed-active conformation even without a ligand bound. Through molecular dynamics simulations, Che et al. revealed how these single mutations alter the DMXAA−STING interaction (DOI: 10.1021/acs.jpcb.6b12472), based on the S162A mutation and a newly discovered E260I mutation endow hSTING AQ with DMXAA sensitivity. On the one hand, the study highlighted that the key-residue arginine 238 dominates the binding of DMXAA and that point mutation can enhance the interaction of R238 with DMXAA. On the other hand, single mutations disturb the coordinated motions of water molecules and alter the amount of water expelled upon ligand binding.

Point 2: Also, the binding of compound 11.

Response 2: Thank you very much for your suggestion, and this is our next step. We are working to resolve the crystal structures of compound 11 with STING and to establish the cellular models of the single mutant STING (162/266), and we hope to elucidate the binding mode of compound 11 in the future. In this paper, we set out to generate 20 poses during the docking of compound 11, which ultimately produced 16 bound conformations (all in the bottom pocket), and we also performed MMGBSA tests on all conformations. We selected the best conformation, according to docking score, glide emodel score, and MMGBSA dG Bind score, to map the binding mode. We take your comment very seriously, and we have improved the docking method and included the scoring of the different conformations in the supplementary information (Table S2).

Point 3: Figure 3 - if possible, please use colors that allow a better contrast between cGAMP and SR717 (green and cyan).

Response 3: It is very sorry that we use similar colors for the different ligands in Figure 3. In the revised manuscript, we have changed SR717 to orange to distinguish it from the other ligands.

Once again, thank you very much for your suggestions. We have studied the comments carefully and tried our best to improve the manuscript, and these changes will not influence the content and framework of the paper. And we marked the changes in red in the revised manuscript we hope to meet with your approval. Please see the attachment for the revised manuscript.

Table S2. Docking scoring of compound 11 with different conformations

conformation

docking score

glide emodel

dG Bind score

1

-6.104

-44.532

-52.36

2

-6.147

-43.547

-32.63

3

-6.274

-43.395

-51.00

4

-5.933

-42.595

-43.86

5

-5.809

-41.833

-48.18

6

-5.869

-40.511

-44.94

7

-5.034

-40.305

-50.88

8

-5.285

-39.637

-31.28

9

-5.679

-39.397

-50.72

10

-4.760

-39.210

-44.73

11

-4.868

-38.995

-35.27

12

-4.734

-38.561

-35.43

13

-5.202

-38.329

-52.32

14

-4.904

-38.087

-34.87

15

-4.777

-37.384

-39.99

16

-4.138

-34.221

-44.66

Reviewer 3 Report

1. At introduction write about docking and cite previous papers related to current studies

1. Scheme 1. COOH group overlapped with benzene ring. Correct it

Over all suitable for publication 

Author Response

We thank you much for providing constructive feedback and admire your rigorous academic attitude. We have completely revised the manuscript (manuscript ID: molecules-2278643) according to your suggestions, which is indeed more reader-friendly. Thank you again for your valuable comments. We have marked the changes in red in the revised manuscript (please see the attachment) and hope to receive your approval.

The correction is as follows:

  1. In the introduction, we have added a portion on docking and cited previous papers related to the current study.
  2. Correction of the error that the COOH group overlaps with the benzene ring in Scheme 1.
